# Experience of Primary Care Nurses of Sexuality Care for Persons with Disabilities: A Qualitative Study

**DOI:** 10.3390/healthcare9121711

**Published:** 2021-12-09

**Authors:** Karim El-Marbouhe-El-Faqyr, María del Mar Jiménez-Lasserrotte, Isabel María Fernández-Medina, Cayetano Fernández-Sola, José Manuel Hernández-Padilla, Laura Helena Antequera-Raynal

**Affiliations:** 1Department of Nursing Physiotherapy and Medicine, University of Almería, 04120 Almería, Spain; eek918@ual.es (K.E.-M.-E.-F.); isabel_medina@ual.es (I.M.F.-M.); cfernan@ual.es (C.F.-S.); j.hernandez-padilla@ual.es (J.M.H.-P.); lar855@ual.es (L.H.A.-R.); 2Facultad de Ciencias de la Salud, Universidad Autónoma de Chile, Terrunco 4780000, Chile; 3Department of Adult, Child and Midwifery, School of Health and Education, Middlesex University, London NW4 4BT, UK

**Keywords:** disabled persons, nursing care, qualitative research, sex counselling, sexuality

## Abstract

(1) Background: Disability is a dynamic interaction between a person’s health conditions and personal and environmental factors. Disability is an evolving concept, which can be improved by intervening in the barriers that prevent disabled people from functioning in their daily life and enjoying a satisfactory sexual life. Sexuality is an important dimension of life that affects people’s well-being. The aim was to describe and understand the experiences of primary care nurses regarding care for the sexuality of persons with disabilities. (2) Methods: A descriptive qualitative study was designed. Twenty-one in-depth interviews were conducted with nurses. A thematic analysis was used to analyse the data. (3) Results: three main themes emerged: (1) Initial assessment of the patient: competencies for a nurse-patient therapeutic relationship; (2) A comprehensive approach to nursing care for persons with disabilities: the importance of sexuality; and (3) Sex counselling in nursing consultations. (4) Conclusions: Nurses have the skills to develop a good therapeutic relationship with patients. Planning of nursing interventions is required in order to promote individual coping, emotional support, and sex education. Sex counselling is essential to promote autonomy, with the figure of the sex therapist emerging for this purpose.

## 1. Introduction

Disability is a public health problem, especially in countries with ageing populations [1]. Worldwide, it is estimated that 15% of people have some kind of disability. Four out of five disabled people live in undeveloped countries [2]. According to the International Classification of Functioning, Disabilities and Health (ICF), disability is a dynamic interaction between a person’s health and personal conditions (motivation, self-esteem) and environmental factors (environment, support and social relationships, services, policies, among others), which represent the circumstances in which a person lives [3]. Disability is a concept that evolves, and persons with disabilities (PD) can develop in their lives if interventions are made to the barriers that prevent them from functioning in their daily life by providing facilitating elements [4,5]. PD can present as a physical disability when individuals suffer from organic abnormalities in the locomotive system or the extremities [6]; a mental disability includes mental retardation, delayed maturation, dementias, and others [7]; sensory and communication disorders include sight, hearing, and language disorders [8]; and multi-disability is when two or more limitations are associated [9]. Limitations may have physical, psychological, and social consequences that affect their work and leisure life and can affect family relationships and social networks. These limitations can become barriers that prevent PD from enjoying a satisfying sex life [10].

A person is considered to have a satisfactory sexual life if they are able to express issues related to sexual life within the framework of values imposed by society [7]. From this perspective, sexuality is how people express and experience themselves as sexual beings [11]. According to the World Health Organization (WHO) [12], sexuality is a fundamental aspect of the human being, encompassing sex, gender identities, sexual orientation, eroticism, pleasure, intimacy, and reproduction. It is experienced and expressed through thoughts, desires, fantasies, behaviours, values, practices, and relationships, among other dimensions; however, not all of them are experienced by one person. It is an important dimension in people’s lives that affects their well-being, filling an affective and relational void [13]; furthermore, sex is essential to have a satisfactory sexual life [14]. Sexuality is influenced by psychological, social, biological, political, cultural, legal, economic, and spiritual factors [12]. Studies suggest that sexuality is ignored as part of the rights of PD [15], others point out that PD have unsatisfied needs related to their sexual and reproductive health [16,17]. In different population groups, there is a prejudice as people believe in false myths that persons with disabilities are asexual beings [15,16,18]. That is, their limitations are incompatible with sexual desire, sexuality, and they cannot have sexual relations [19]. They are also hampered by social barriers, which imply an inability to develop romantic and sexual relationships, which limit opportunities to interact with other partners [18]. However, different studies show that a similar percentage of persons with disabilities and non-disabled people are married and are alike in various aspects of sexuality and reproduction [19].

Ignorance of the sexual needs and sexual health of persons with disabilities becomes a limitation for them to have access to resources and information [14]. This implies that, in practice, when public policies are created for these sectors of the population, their sexuality is hardly taken into account, and there is a lack of specialised multidisciplinary teams [16,20], in addition to there being less access to information on reproductive health and advice on sexual health [21]. Nursing, as an integral part of the health care system, works towards the prevention of diseases, towards the promotion of health and the care for people with physical or mental illnesses or with disabilities, at all ages in the health care system [22]. The nursing staff are greatly valued by PD, and their direct contact in primary care consultations results in the creation of a bond of trust that helps them to understand PD [23]. From this perspective, it is possible to observe the need to pay greater attention to sexual needs, beginning with the nursing consultations in Primary Health Care. This demand comes from society, and from PD themselves [24]. Within the assistance given to PD, the sexual aspect has been ignored as it has been considered a taboo subject [19]. PD may have all their needs satisfied but be limited in the sexual sphere due to not having the optimal capabilities to perform it, giving rise to the figure of the sex therapist [25]. Sex therapy aims to help PD so that they can achieve full sexual satisfaction, increase their self-esteem, and develop their autonomy so they will be empowered [26,27]. In countries such as Switzerland, Denmark and Belgium there are regulations regarding the profession of the sex therapist [28].

Previous studies have analysed epidemiological aspects of patients with disabilities, designed instruments that facilitate the assessment of sexuality [29], increasing the protection of their rights in their development [15], and pointed out the lack of access to sexual and reproductive health services [16,30], but few studies have explored the experiences of nurses in caring for sexuality in PD [10,31]. Therefore, studies are needed that examine the health care performed by Primary Care nurses in the development of competencies to improve the attention to the sexual needs of PD, thereby promoting their autonomy. Our research question is: what are the experiences of primary care nurses regarding care for the sexuality of persons with disabilities? The aim of our study is therefore to describe and understand the experiences of primary care nurses regarding sexuality care for persons with disabilities (PD).

## 2. Materials and Methods

### 2.1. Design of the Study

This is a descriptive qualitative study that allows an in-depth description, from a naturalistic perspective, of a little-known phenomenon by exploring the experiences of the protagonists [32]. This approach is suitable for exploring and gaining insight into experiences in health settings from the perspective of health professionals [33]. The Consolidated Criteria for Reporting Qualitative Research (COREQ) [34] were followed.

### 2.2. Participants and Environment

The study took place in the consultation rooms of several Primary Health Care centres. Participants were selected using an intentional sampling technique, fulfilling the following inclusion criteria: being a nurse in Primary Health Care, experience in treating PD, and having given informed consent. The only exclusion criterion was refusing to participate in the study. For the recruitment of the participants, one of the researchers (K.E.-M.-E.-F.) invited the professionals to participate by means of a phone call, and an appointment was made. Twenty-four nursing professionals were invited to participate, and three declined due to lack of time. A total of 21 nurses were interviewed. The sociodemographic characteristics of the participants are shown in Table 1.

### 2.3. Data Collection

The data collection included 21 in-depth interviews (IDIs) developed between the months of January and June 2021. The interviews were carried out in the nursing consultation rooms of the health centres where the nurses carry out their work. Each participant gave only one individual interview, with an average duration of 43 min. The IDIs were carried out by three researchers trained in qualitative research (K.E.-M.-E.-F., M.d.M.J.-L., L.H.A.-R.), following a semi-structured interview guide with relevant questions (Table 2). Before starting the IDIs, sociodemographic data was collected, the protocol was explained, data confidentiality was guaranteed, and consent was signed. The interviews were in Spanish, recorded on audio and transcribed word-for-word. Afterwards, the participants had the opportunity to read the transcripts to verify the content. Data collection ceased when data saturation was reached, that is, when no new information emerged from the data.

### 2.4. Data Analysis

The interviews were transcribed and analysed with the qualitative analysis software Atlas.ti 9.3, by three independent researchers (K.E.-M.-E.-F., M.d.M.J.-L., and I.M.F.-M.), together with the annotations of the researchers. The thematic analysis was carried out following the phases described by Braun and Clarke [35]: (1) Familiarisation with the data consisted of a complete reading of all the transcripts and a rereading for early knowledge of the experiences and taking notes of initial ideas; (2) Generation of the initial codes: the data groups were systematically codified; (3) Search for themes: the codes were associated and grouped into themes with patterns of shared meaning with a central idea; (4) Review of topics: the topics were checked to make sure they were consistent with the codes and their consequent results in subtopics and topics; (5) Definition and nomenclature of topics: the details of each topic were analysed and refined. (6) Preparing the report: examples of topics and subtopics were selected, the analysis was related to the research question, and a final report was made (Table 3).

### 2.5. Rigour

To ensure rigour, the criteria proposed by Guba and Lincoln [36] were adapted. Credibility: the data collection process was detailed. The process was helped by the triangulation of researchers. It was carried out by the researchers independently, and the data was later pooled. The researchers were nurses, with extensive experience in the healthcare processes. Transferability: Detailed information was provided about the participants, the surroundings of the study, the context, and the method. Reliability and confirmability: the researchers made the transcripts, which were reviewed by other members of the research team. Finally, the verbatim transcripts of the participants’ experiences were incorporated into our results by means of citations, verified by the participants, and this was able to contribute to the rigour of our study.

### 2.6. Ethical Considerations

All registrations are carried out respecting the precepts established in the current legislation on the protection of personal data contained in Organic Law 3/2018 of 5 December on the protection of personal data and guarantee of digital rights; as well as in Regulation (EU) 2016/679 of the European Parliament and of the Council of 27 April 2016, and Law 41/2002 of 14 November, the basic regulator of the Autonomy of the Patient and Rights and Obligations regarding Information and Clinical Documentation. The participants were informed of the purpose of the study and the voluntary nature of their participation. Before starting the study, informed consent was collected, and permission was requested to record the interview. This study was carried out taking into account the ethical principles of the Declaration of Helsinki. This research obtained the authorisation of the Ethics Committee of the University of Almería (EFM-64/18).

## 3. Results

All participants identified themselves as nurses. Of those, 33.33% were men and 66.66% women. The mean age of the participants was 42.24 years. The average period of professional experience was nine years. Three topics and seven subtopics were extracted from the inductive analysis of the data. All of them help us understand the experience of primary care nurses regarding sexuality care for persons with disabilities (Table 4).

### 3.1. Theme 1. Initial Assessment of the Patient: Competences for a Nurse–Patient Therapeutic Relationship

The nursing assessment is a planned, continuous, and systematic process focused on obtaining all the information about the health status, in order to provide comprehensive care for the patient. This topic reflects the experiences of nurses in the PD assessment phase. The nurses relied on active listening as a communication technique in order to conduct a good interview. They emphasised the importance of this technique together with the use of a simple and appropriate language for the patient, which favoured the creation of an environment of trust to deal with issues as intimate as identity, orientation and sexual needs of PD.

#### 3.1.1. Subtheme 1.1. The Clinical Interview: Starting Point for Good Communication

The results of the study showed that the clinical interview is one of the most important methods in order to get to know the opinions, expectations, and experiences of patients. For the nurses to deal with such an intimate issue as sexuality, it was essential that the PD played an active role and participated in decision-making. For this reason, the nurses indicated active listening, empathy, and communication skills. Knowing how to listen meant understanding what the patient was saying and, in turn, observing their non-verbal language such as posture, look, or tone of voice. For the participants, active listening improved interpersonal communication with the patient, the key to creating a trusting relationship where the patient was relaxed and could express their feelings and wishes:


*“The patient must feel comfortable to open up to the nurse, many come with anxiety, shame … they will never tell us what their problems are if we can’t get them to relax and trust us”*
(Participant 6)

The participants stressed that it was essential to maintain eye contact with the patient. It was important to show non-verbal behaviour that communicated their commitment, and, above all, it was necessary to avoid interrupting the patient at key moments when they were talking. Active listening required listening carefully, with empathy, accepting the ideas, opinions and feelings of the person. As one of the nurses pointed out, to address patients’ problems, you had to understand their needs without judging what they said:


*“A patient who you don’t look at, who you don’t pay attention to, and who you don’t listen to, is never going to open their heart up to you. The nurse who judges patients to have some type of disability will not be able to make advances in the care. The best way to address these needs is to do things with respect and empathy”*
(Participant 13)

According to the participants, the information given to the patients had to be clear, easy, simple, concrete and comprehensible, using simple language. The first interview was the most important, and the information had to be adapted to the cognitive capacity and the needs of the PD. As one of the nurses in our study highlighted:


*“There are interviews where the patient comes with a lot of information from home and really knows what they need, others report that the information they have about sexual needs is generic and does not solve their problems … we try to give them support to cover their deficiencies in addressing sexual issues, without burdening them with unnecessary medical technicalities”*
(Participant 9)

#### 3.1.2. Subtheme 1.2. Transmitting Security and Confidence in Approaching Sexual Needs

The possibility of a patient attending a nursing consultation with the confidence of being able to consult on any sexual problem or question was key for the participants. In order to know the sexual needs of the PD, the nurse had to know the anatomical, physiological, biological, medical, and psychological factors that have a great impact on their intimate lives. The participants therefore emphasised that it was essential to foster a climate of trust, calm, intimacy and security to interview PD individually; some were introverted and others more receptive to this type of intervention. As one of the participants pointed out when she managed to make the patient receptive and secure about their intimate life, the couple should be included in the therapy:


*“Many times my patients tell me that when they leave the consultation they try to practice the concepts they have learned, and that little by little they are breaking the chains that keep them away from sexual enjoyment”*
(Participant 3)


*“The main aim of the nursing interview is for the patient to go out the door with enough tools to face their sexual life in a full and secure way”*
(Participant 15)

The participants in the study reported that once the patient felt comfortable and trusted their nurse, communication about their sexual experiences and needs was facilitated. PD could describe their experience of the sexual act if they were able to carry out intercourse without any type of limitation; if they felt satisfied with intercourse, or whether there really were sexual limitations. As several of the participants pointed out, the type of disability can influence the development of their sexuality, and for this reason questions were asked about their genitality and intercourse, but also their affectivity, self-esteem, and other abilities. All these factors determine whether the development of their affective-sexual relationships is satisfactory or whether there is some sexual limitation:


*“The problems that underlie their basic problem, together with emotional problems and difficulties in finding a partner, contribute to a less active and satisfying sexual life”*
(Participant 1)


*“Sexual dissatisfaction in patients with disabilities is related to personal, social, and medical factors. Difficulty to find a suitable position during intercourse can cause stress in the couple, and this would inevitably lead to an incomplete sexual relationship”*
(Participant 18)

#### 3.1.3. Subtheme 1.3. Functional Diversity: Exclusion of Gender Identity and Sexual Orientation

Starting from the idea that there are as many sexualities as there are people, the participants in this study pointed out that PD are wrongly treated as being asexual, and they are also infantilised. Thus, the nurses encountered obstacles such as these false beliefs when addressing sexuality in PD, making it difficult for them to live their own sexuality alone or as a couple. As one of the nurses pointed out, it was necessary to talk about sexuality and sex normally, and for them to feel free to live it:


*“Sexuality should be considered as another basic need, without shame, openly; whoever has a partner should experience and include new techniques, while those who are single should discover new tools to meet this need”*
(Participant 3)


*“The nurse must have an open mind to these sexual practices and must break sexual taboos since our society is changing, and so they must change the mentality of health personnel”*
(Participant 21)

PD were also accompanied by a distorted vision, with a binary sex-gender scheme. However, some of the PD identified themselves within the LGTBQI + community. For the participants, knowing the gender identity and sexual orientation of the patients was essential to be able to efficiently address their sexual needs. Sometimes, when PD were attended to, just the disability was taken into account as their only characteristic, ignoring other characteristics such as gender identity. As one nurse pointed out:


*“Some time ago, ago, as I had never asked my patient about this, I talked to him about the female reproductive system, and in the end it turned out that he was homosexual”*
(Participant 12)

During the interview, the participants tried to dispel the doubts that PD might have, promoting their acceptance, integration in life and their self-esteem so that they could achieve a satisfactory sexual life. Knowing the gender identity of the patient was essential to prepare a good interview, using an adapted professional language. As several participants pointed out, in the initial contact the first impressions were created that served to build a good therapeutic relationship:


*“Today multisexuality is rooted in our society, and nursing must be up-to-date in this type of practice since an outdated care plan is not going to achieve its goal”*
(Participant 5)


*“The care plan that we prepare for the patient must contain yes or yes, the sexual identity of the patient. In this way we will be able to reinforce the positive points and help them to overcome their fears regarding sexual practice according to their sexual inclination”*
(Participant 14)

### 3.2. Theme 2: Comprehensive Approach to Nursing Care for Persons with Disabilities: The Importance of Sexuality

A good preparation of all the planned interventions and activities was the key to being able to achieve the objectives set with the patient and to solve the problems related to their sexuality. This topic shows the nursing care provided to PD in order to promote individual coping and education in sexuality as the culmination of this process.

#### 3.2.1. Subtheme 2.1. The Process of Coping in Persons with Disabilities: Emotional and Partner Support

For the nurses, effective coping consisted of the efforts of PD to manage and reduce internal and environmental demands. The problems they suffered and the treatment they underwent required efforts to overcome. Physical and biological aspects influence sexuality, and for this reason the participants pointed out that it was essential to promote interventions that address health, including sexual health. Nursing care was focused on promoting their ability to cope so that they would find stability between the problem, the treatment, their emotional state, and their limitations, as indicated by several participants:


*“In the consultation, the aim is to ensure that the patient is able to find the necessary tools to strengthen their personal capacity, connect with their partner, overcome the psychic-physical limitations that their disability implies so they can develop and enjoy their sexuality”*
(Participant 13)


*“I am amazed at the ability that some patients have to get ahead, I have patients who cope with life with an energy and vitality that makes me doubt whether I am really helping them or are they helping me”*
(Participant 10)

The emotional support helped them to improve and face the problem to be solved. The participants in this study showed themselves to be the key professionals to give the emotional support that PD, their partner, or family needed. In the consultation they created a listening space, where the patient could feel safe and accepted, through an open and empathetic dialogue. One nurse pointed out that during the nursing interview, she tried to encourage the expression of feelings and thoughts about the situations that concerned them. The aim of the interview was to offer real and accessible resources that would allow PD to face this new situation, counteracting negative thoughts and emotions. The intervention was focused on enhancing body image and self-esteem, which, as indicated by several nurses, are aspects that have usually been affected, and which can cause emotional disorders and alterations in sexuality.


*“Emotional therapy is the fundamental pillar of our intervention, the patient leaves strengthened and gains the capacity to solve and attend to all their sexual needs”*
(Participant 11)


*“For every woman or man it is important to feel attractive, to your partner or everyone else. That you look good, that you feel desired … that is important for self-esteem, and therefore it is something that we must work towards right from the consultation with the patients”*
(Participant 3)

Another fundamental aspect in the work on the sexuality of PD were relationships, personal relationships and the relationship with their surroundings. PD had difficulties interacting with other people or experienced situations where they lacked sexual intimacy, and this may have been caused by family overprotection, lack of social contacts, or the degree of disability. Emotional support was one of the professional skills that contributed to nursing care. The participants tried to reach the PD through emotional support, providing tools that could resolve the lack of intimacy with their partner. With this emotional support, the intention was to create a bidirectional channel that implied an emotional exchange between the patient and their partner; where each revealed their own feelings and fears:


*“The moment that our patient opens up to their partner and reveals what their fears are and sees that their partner is helping them to overcome them is the most rewarding thing in our profession”*
(Participant 15)


*“Patients with a stable partner are those who often demand more information and want to know how to increase pleasure in their intimate relationships”*
(Participant 1)

#### 3.2.2. Subtheme 2.2. Sex Education for Persons with Disabilities: Nursing as an Agent of Health Education

In Primary Health Care, a community setting where the nurses in this study had experience, they developed health education to learn healthy lifestyle habits. It was important to carry out such health education with the PD so that they knew the specific needs they had at the individual, family, and community level, as well as assessing what help they needed to achieve sexual satisfaction. Nurses were prepared to be the main agent of health education, as one nurse highlighted: *“You must reinforce and help the patient to learn to make autonomous and responsible decisions, giving them knowledge and tools to solve health problems that may appear together with their disability”* (Participant 20). Participants informed, motivated, and helped PD to improve their practices and lifestyles:


*“The important thing about our work is that the patient breaks that psychological barrier that prevents them from enjoying their sexual relations. I always instil in them the idea that to overcome their fears they must be able to get out of their comfort zone and seek their sexual satisfaction”*
(Participant 15)


*“In the consultation we attend patients with acquired disabilities such as hemiplegic patients after a traffic accident, who go from having a full sexual life to facing a new situation where the couple sometimes feels guilty, or there are problems with getting excited. We have to carry out sex education work with the patient and also with the partner and family!”*
(Participant 17)

One of the fundamental values of health education is prevention. In this sense, health education should be focused on offering information on attitudes towards sexuality so that they have safe sex, informing them of the existing preventive methods and which are the contraceptives that they could use according to their personal characteristics. As several of the participants pointed out, it was very important to identify the risk factors that PD had and even situations of abuse, in order to give them the information, the necessary skills, and required support so they could protect themselves from risky practices, both in masturbation and intercourse:


*“I work with patients with Down’s syndrome, and from my professional experience we have to always be vigilant because it has already happened to us that she has left with a partner to have sex on open ground; this is dangerous, and they can get infections …”*
(Participant 19)


*“We have to know how to combine the right to sexuality with sex education; they have to know how to distinguish a normal relationship from another that is abusive”*
(Participant 10)

### 3.3. Theme 3: Sex Counselling in Nursing Consultations

Sexuality is a fundamental aspect of the human being throughout their life, and any alteration in its development can directly affect the well-being of the patient. Sexuality is not a neutral phenomenon as each person lives and experiences it in a unique way. Thus, in the nursing consultation we must address sexuality in PD from a physiological, psychological, and also social perspective. This topic describes the importance of sex counselling from a comprehensive perspective, including physical, intellectual, emotional, and social needs, promoting the autonomy of the patient. In addition, the figure of the sex therapist is described as a support when PD are not self-sufficient and depend on third parties to satisfy their sexual needs.

#### 3.3.1. Subtheme 3.1. The Approach to Sexual Behaviour in Persons with Disabilities

Sexuality is part of integral health, and on many occasions, PD have been deprived of satisfying their sexual needs. Sexuality is a process that implies affection, attachment, the need to feel, to have contact, intimacy, feel pleasure and love. Any alteration in physical, emotional, and social development can influence the personal relationships of PD and alter their sexual development. Beginning with the nursing consultations, the participants in this study worked to improve the physical and emotional capacity of the PD, increasing their self-confidence. The nurses emphasised that it was necessary to increase knowledge, improve skills, and for PD to learn to know each other and explore their eroticism. This helped reduce their anxiety and embarrassment. One nurse pointed out that the best way to help PD to improve their sexual behaviour was through appropriate counselling:


*“Good care at the sexual level can help to increase the well-being and personal happiness of the patient. Until recently talking about sexual health was considered a taboo subject”*
(Participant 8)

Having a healthy sexuality meant that PD were happy and self-confident. With health counselling, the participants tried to redirect patients’ risky behaviour, to teach them to explore their eroticism, and help them to develop tools that would improve their individual and their partner’s sexuality. As one nurse noted: “We explained to our patients that sexuality includes a general whole and is not simply the absence of disease or dysfunction” (Participant 13). For our participants, adequate counselling consisted of working on communication, sexuality, social skills, and not just reproductive problems and sexually transmitted diseases:


*“All people should know about sexuality and especially this type of patients who, in addition to their disability, can bring other family or social deficiencies that are more difficult to overcome than the limitation itself”*
(Participant 2)


*“We cannot base sexuality only on genitality, it is important to also focus on their own experiences, on their own body, and make a sexual adaptation to offer them alternatives. Encouraging kisses, caresses, eroticism … which will also give them pleasure”*
(Participant 17)


*“I work with patients with intellectual disabilities, and all women attend their medical follow-ups, specifically gynaecology, especially on the question of contraceptives to avoid unwanted pregnancies, in addition to avoiding STDs…”*
(Participant 18)

#### 3.3.2. Subtheme 3.2. The Figure of the Sex Therapist

The results of this study pointed out that sex therapy, as well as the figure of the sex therapist, is essential for all PD, with great needs of support and to be able to satisfy their sexual needs. The PD, depending on their limitations, required help to eat, to clean themselves, or other basic daily activities. The sexual needs of people should be understood as a complement to the personality and the human being, and the sex therapist can be seen as a figure analogous to the personal assistant or home help. As several of the participants noted:


*“Sex therapy is understood as the help given to a patient so that they can fulfil their sexual needs”*
(Participant 7)


*“The sex therapist is the professional who provides sexual-sensory therapy to patients with disabilities”*
(Participant 5)

The figure of the sex therapist has to be framed in the process of care for the patient; although such sex therapy is carried out by people outside the healthcare world. The nurses highlighted that the figure of the sex therapist was novel and unknown, and therefore it is important that it be controlled and monitored by a health professional. Although this figure is not recognised in many countries, they highlighted the existence of this figure in order to deal with the sexual needs of PD. The sex therapist needs to have specific training, as stressed by a nurse: “*Sex therapists must be specifically trained in sexuality in the context of disability, they will not only be facilitators of the sexual act that the PD cannot carry out on their own, but they will also help them to discover their own body, to regain their self-esteem, and even to advise their own family*” (Participant 4). For our participants, it was also important for health personnel to be trained in new ways of approaching sexuality in PD:


*“It is incomprehensible that we have to train ourselves in these therapies, when it is the health service that should provide them”*
(Participant 9)

The figure of the sex therapist also had to be regulated and protected by law since it became necessary, and in this way confusion with covert prostitution was avoided.


*“The only way for it not to be confused with prostitution is the need for training to ensure the professionalisation of the figure of the sex therapist”*
(Participant 6)


*“No, we cannot confuse it with prostitution. They are providing a service, attending to the specific needs of persons with disabilities”*
(Participant 20)

For the participants, it was important to fight for true equality, but they lacked the right tools to do so, this being a failure on the part of legislators. Society advances, modernises, and requires the normalisation of all the activities that could cover the basic needs of PD, in order to make visible the right of these people to achieve a full and dignified life.


*“Although it is not recognised in all countries, it is a fact that the figure of the sex therapist exists. We still live in a society that discriminates against persons with disabilities, and, faced with this, we must fight for society to advance and find a way to meet their needs”*
(Participant 4)

## 4. Discussion

The aim of this study was to describe and understand the experiences of primary care nurses regarding sexuality care for persons with disabilities (PD). Previous studies show a lack of sexual health coverage for persons with disabilities, which may be indicative of the stereotypical belief that persons with disabilities are asexual beings [15,37] or are unable to develop romantic and sexual relationships [32]. Knowing the experiences of nurses can improve our knowledge of sexuality in PD, extending needs and knowledge to also include affective and sexual health, in order to move away from the traditional idea that desexualises persons with disabilities [16].

According to the ICF [3], disability is a dynamic interaction between a person’s health conditions and personal and environmental factors. PD can improve if they are provided with facilitating elements that help them to function in their daily life [4]. The results of this study show that it is essential for the patient to play an active role and participate in decision-making [20,38], and the clinical interview is one of the most important methods to discover the sexual experiences, identity, or needs of patients [39]. It is essential that, if attention is given to understanding sexual needs, it must be coherent and adapted to the limitations that can influence PD [40]. Techniques such as active listening, empathy, and using adapted language, which is accessible to the needs of persons with disabilities [14,41], will improve communication and also help to create a therapeutic nurse–patient relationship [37,41]. In Primary Care consultations, the nurses have direct contact with the PD, and create a climate of trust where patients and their partners feel free to express their fears, needs, or doubts about sexual aspects that have arisen in their care process [23,40]. As with other groups such as young people, it is very useful to offer a safe and protective environment to be able to explore their identity, orientation, or sexual behaviour [42]. Consistent with other studies, the results show that PD identity and sexual orientation are often ignored, interpreting care for their disability as their only requirement [30].

As defined by the WHO [12], sexuality is a fundamental aspect of the human being, that includes sex, gender identity, sexual orientation, eroticism, pleasure, intimacy, and reproduction. That sexuality has implications for all spheres of the human being; and by not allowing its healthy development, it interferes with a dignified, quality life, affecting the development of identity, a life project, and the exercise of rights, duties, and participation in social life [40]. Limitations of the degree of disability, family overprotection [43], illnesses, or drug therapies [41] interfere in the possibilities for PD to have interpersonal relationships, as well as in the development of their sexual intimacy. The results of this study suggest that the care provided by nurses should be focused on increasing personal coping in order to find stability between the patients’ limitations and their sexual sphere. Like Ferri [41], we agree that the sexual sphere is not the first problem that PD have to face, but it is a fundamental aspect that must be solved in order to satisfy sexual needs. The perception of their own image can cause internal conflicts within the self that influence the sexual sphere [41]. Self-esteem and body image must be improved in order to increase self-confidence, and with this their personal development [44], so that they become sexually healthy people and maintain social relationships [43]. As Navarro-Martínez [45] points out, nursing interventions help reduce the emotional burden of patients and their families. Along the same lines, the emotional support provided by the participants contributes to the comprehensive care of PD. In the consultation, efforts should be made to equip PD with instruments or individual tools focused on the person and their emotions [46], in order to learn to maintain autoerotic behaviour in private and to control emotions with their partner [43,47]. For the full development of this therapeutic activity, it is necessary for nursing consultations to plan sexuality education and work on this during nursing visits [14,45]. Sexuality education would help PD to satisfy sexual needs and to make decisions with the necessary information in terms of responsible and safe sexual relations [45]. Previous studies show a lack of interventions of health services related to reproductive health and protection against sexually transmitted diseases [19], promoting the use of specific resources and technological advances that allow the creation of new strategies to promote nursing care [14].

The participants in this study worked to promote PD self-esteem, their knowledge, improving personal and social skills, with the aim that the patient learns to explore their sexuality. We agree with other studies that show that persons with disabilities are less likely to participate in social activities, and therefore it is important for a marital relationship to have this social support [48]. This lack of support can also be seen in health care as sociocultural issues resulting in the fact that persons with disabilities do not receive counselling about sexuality as some professionals consider it a private matter [10]. In terms of the importance of sex counselling, we agree with other studies that highlight the importance of sex education for patients but also for the partner, family and even the caregiver as a basis for PD to maintain appropriate sexual behaviour and avoid risky behaviour [49]. In addition, a new figure has emerged, that of the sex therapist, who will try to respond to the specific needs of PD. This figure is understood as an advisor who carries out or helps PD to carry out daily life tasks because they cannot carry them out by themselves [26], tasks that will be previously agreed upon by a contract between the PD and their assistant [24]. Therefore, the sex therapist becomes a guide who ensures that PD learns to know their own body, to enjoy themselves, offering advice and helping them to overcome their problems [50], as well as increasing empowerment so that they can manage their own selves [51]. The results of this study show the need for specific training for sex therapists in sexuality in PD [27,31], on the different types of disability, and, on the other hand, carrying out sex education in work on eroticism [26]. Finally, the participants of this study agree that the figure of the sex therapist is framed within a role of helping PD and has no similarities with prostitution [31]. We cannot confuse this type of assistance with that of being an active part of the sexual relationship [28]. However, authors [26] defend the fact that the sex therapist can help the PD to carry out sexual practice.

### Strengths and Limitations

Our results expand on those of prior qualitative studies referring to sexuality and persons with disabilities. The participants came from different health care centres, assuring representation of diverse points of views and the transferability of the findings. As limitations of this study, firstly the results are influenced by the sociocultural factors of the nurses of the health system where the study is framed. To compare the results, similar studies could be carried out in other countries. Secondly, this study only reflects the experiences and perspectives of health professionals, in the figure of the sex therapist. For future studies, it would be interesting to include the perspective of people who have experience as sex therapists for people with sexual disabilities as well as to include the perspective of the target population and their families, in order to obtain a deeper understanding of the phenomenon.

## 5. Conclusions

The results of this study suggest that nurses have the skills to develop a good therapeutic relationship with PD. For this, they rely on different communication techniques such as active listening, empathy, or the use of simple language to be able to carry out a good initial assessment interview of the patient. This fosters an environment of trust that helps to deal with elements related to basic needs, gender identity, or aspects of sexuality. Good planning of nursing interventions is required to provide adequate attention to the sexual needs of persons with disabilities, taking into account physical, psychological, and social aspects. Nursing care aims to increase individual coping, emotional support, and sex education for the patient, their partner and their family. Sex counselling is essential to address the needs of these patients from a comprehensive perspective and to promote their autonomy. In a nursing counselling framework, the figure of the sex therapist emerges as a new element to consider in order to introduce improvements in the sexuality of PD.

## Figures and Tables

**Table 1 healthcare-09-01711-t001:** Sociodemographic characteristics of the participants (N = 21).

Participant	Gender	Age	Country	Years of Experience	Place of Work
1	Female	30	Spain	4	AHS
2	Female	37	Spain	9	AHS
3	Female	45	Spain	10	AHS
4	Female	53	Spain	10	AHS
5	Female	31	Spain	4	AHS
6	Female	38	Spain	10	AHS
7	Male	42	Spain	10	AHS
8	Male	57	Spain	11	AHS
9	Male	36	Spain	5	AHS
10	Male	27	Spain	4	AHS
11	Female	38	Spain	9	AHS
12	Male	40	Spain	10	AHS
13	Female	55	Spain	12	AHS
14	Male	39	Spain	10	AHS
15	Female	58	Spain	13	AHS
16	Male	47	Spain	10	AHS
17	Female	43	Spain	10	AHS
18	Female	48	Spain	12	AHS
19	Female	53	Spain	11	AHS
20	Female	34	Spain	6	AHS
21	Female	36	Spain	7	AHS

AHS: Andalusian Health Service.

**Table 2 healthcare-09-01711-t002:** Interview protocol.

Subject	Content/Possible Questions
My intention	Learn about the experiences of nurses regarding care for the sexuality of persons with disabilities.
Ethical issues	Inform that participation is voluntary, registration, consent, data confidentiality, and the possibility of leaving the study at any time.
Introductory question	Could you tell me about your experience in the nursing assessment of a patient with a disability?
Conversation guide	How do you approach the clinical interview to discover the sexual needs of the disabled patient? How do you create a climate of trust? Tell me about how you carry out health education on sexuality with people with disabilities. Could you explain to me what sex therapy is? Could you describe the figure of the sex therapist to me?
Final question	Is there anything else you would like to tell me?
Appreciation	Thank them for their participation, remind them that their interview will be of great use, and place ourselves at their disposition.

**Table 3 healthcare-09-01711-t003:** Example of the codification process.

Quote	Units of Meaning	Subtheme	Theme
“*For every woman or man it is important to feel attractive, to your partner or everyone else. That you look good, that you feel desired … that is important for self-esteem, and therefore it is something that we must work towards right from the consultation with the patients*” (Participant 3). “*Patients with a stable partner are those who often demand more information and want to know how to increase pleasure in their intimate relationships*” (Participant 1).	Illness, treatment, body image, self-esteem, intimacy, anxiety.	Coping process in people with disabilities: emotional and partner support.	Theme 2. Comprehensive approach to nursing care for people with disabilities: importance of sexuality
*“You must reinforce and help the patient to learn to make autonomous and responsible decisions, giving them knowledge and tools to solve health problems that may appear together with their disability”* (Participant 20). *“We have to know how to combine the right to sexuality with sex education; they have to know how to distinguish a normal relationship from another that is abusive”* (Participant 10).	Safe sex, preventive methods, autonomy, specific needs	Sex education for people with disabilities: nursing as a health education agent.	

**Table 4 healthcare-09-01711-t004:** Themes, subthemes, and units of meaning.

Theme	Subtheme	Units of Meaning
1. Initial assessment of the patient: competences for a nurse–patient therapeutic relationship	1.1. Clinical interview: starting point for good communication.	Empathy, not judging, eye contact, not interrupting, receptive, fears, comprehensible, trusting environment.
	1.2. Transmitting security and confidence in addressing sexual needs.	Comfort, sexual activity, sexual arousal, masturbation, intercourse, satisfaction, sexual limitations.
	1.3. Functional diversity: exclusion of gender identity and sexual orientation.	Acceptance, integration in life, self-esteem, sexual identity, gender identity, multisexuality.
2. Comprehensive approach to nursing care for people with disabilities: importance of sexuality.	2.1. Coping process in people with disabilities: emotional and partner support.	Illness, treatment, body image, self-esteem, intimacy, anxiety.
	2.2. Sex education for people with disabilities: nursing as a health education agent.	Safe sex, preventive methods, autonomy, specific needs.
3. Sex counselling in nursing consultations	3.1. Approach to sexual behaviour in people with disabilities.	Intimacy, confidentiality, anxiety, shame, enhance self-esteem.
	3.2. The figure of the sex therapist.	Unknown, training, regulated, therapist, prostitution.

## Data Availability

For confidentiality purposes, the data is in the possession of the author (K.E.-M.-E.-F.).

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
