# Peer review of "Experience of Primary Care Nurses of Sexuality Care for Persons with Disabilities: A Qualitative Study"

_healthcare, 2021, doi:10.3390/healthcare9121711_

Round 1

Reviewer 1 Report

Here are some comments and suggestions:

  • Keywords should be MeSH terms.
  • The introduction is complete and relevant to justify the study
  • The methodology has been explained clearly and coherently and responds to the objective of the proposed study. In addition, rigorous methodological strategies have been included to clarify the criteria used for the analysis.
  • It would be interesting to know which author or authors of reference were used for the analysis from a naturalistic paradigm. A description of the theoretical concepts would allow to complete the discussion and achieve a greater degree of abstraction in the data analysis.
  • The results respond to the analysis of the verbatims included. Relevant results have been raised, accompanied by living codes consistent with defined categories and codes.
  • It would have been interesting to include an outline of topics and subtopics and their relationships.
  • The discussion is adequate and the main limitations of the study are included.

Author Response

Thank you for the comments. I add a document with the review. 

Reviewer 2 Report

The article is well-organized, well-written, and easy to understand; discusses a very interesting subject, often taboo and often ignored. It is therefore a very relevant theme. Treats the responsibility of primary care nurses in promoting sexuality of people with disabilities and allows to conclude that it is necessary to the planning of nursing interventions, to promote individual confrontation, emotional support, and sexual education, as well as sexual counseling, is essential for promoting autonomy by pointing to the need to exist a sex therapist that can provide help and support for these people.

The abstract: It is a single paragraph, but the structure is not clear. We can’t understand where the end of the results is and where is the beginning of the conclusions if there are conclusions in these abstract…this is not clear. Please follow the suggested structure present at https://www.mdpi.com/journal/healthcare/instructions#manuscript

  • Background: 2) Methods: 3) Results: 4) Conclusion:

The introduction: The introduction puts the study in a broad context and justifies why it is relevant. They define the purpose of the study and its significance, including the “state of art” of the research field and key publications were cited. The authors mention the aim of the work and I suggest that they briefly highlight the main conclusions.

Materials and Methods: the methodology is clearly explained. In fact, the authors describe the method in detail that allows other researchers to replicate the study. The study design, the participants, data collection, rigor, and ethical aspects are well described. They give the name and version of the software used (Atlas.ti 9.3) and include a very good interview protocol. I will make some brief suggestions: The sample is in the quantitative paradigm and its concept means that it is a part that represents the population. Therefore, by its nature and in good rigor, this concept should not apply to qualitative studies, and I suggest replacing the term sample by participants/informants and, please, clarify how you did the data triangulation (line 142). They respect the COREQ criteria.

Results: The authors provide a well-made concise description of the results, highlighting themes and subtitles and illustrating them with the relevant speech of the informants The findings are relevant to understanding the phenomenon under study.

Discussion:  the discussion is combined with findings. Authors discuss the findings and contrast their findings with previous studies. Study limitations and suggestions are highlighted as well as future research directions are pointed out.

Conclusions: The authors answered the research question and summarized the findings. The relevance of the study, by understanding the theme of sexuality in people with disabilities, and understanding how nurses and therapists can help these people, are quite underlined here.

References: Less than 50% of references are less than 5 years old.

Some more suggestions: Lines 13 e 14: I suggest a better articulation between these two ideas Disability is a dynamic interaction between a person's health conditions and personal and environmental factors that represent the circumstances in which they live, and These limitations can become barriers to enjoying a satisfying sex life.

Line18: Please consider including the type of interview here.

Line 21: (3) Sex counselling in nursing consultations. I think maybe you want to end with a colon (:)

Author Response

Thank you for your comments. I add a document with the review. 
